# Determinants of Financial Performance in China's Intelligent Manufacturing Industry: Innovation and Liquidity

**Guanghong Zhang** [1] and **Yune Lee** [2,*]

1 Department of International Trade, Graduate School of Humanities and Social Sciences, Incheon National University, Incheon 22012, Korea; cherryzhang1314@hotmail.com
2 Department of International Trade, Incheon National University, Incheon 22012, Korea
* Correspondence: leeyune@inu.ac.kr; Tel.: +82-03-2835-8543

**Abstract:** This study focuses on the mediation channels through which the financial performance of intelligent manufacturing industries closely related to the Fourth Industrial Revolution has been affected. Along with compiling a massive volume of datasets publicized by the Chinese government and other authoritative institutions, a survey of the 317 listed enterprises of the intelligent manufacturing industries in China has been established for statistical analysis. Using Structural Equation Modeling (SEM), this research tests six hypotheses and confirms the inter-factor impact relationship between exogenous and endogenous factors. We find that innovation efforts mainly led by increasing investment in Research & Development (R&D), along with high liquidity, surely lead to good financial performance, whereas innovation efforts alone do not. Government support policy has been found to be closely related not only to higher liquidity, but to good financial performance through the common channel of R&D investment. Regional innovation capability has been revealed to be related to R&D investments, and, furthermore, to liquidity, which shows that the regional innovation system in China has been functioning relatively well to induce enterprises to increase investments and secure higher liquidity, and finally contribute to achieving better business performance. However, regional economic development shows no relationship with R&D investments, and consequently neither with liquidity nor with performance.

**Keywords:** intelligent manufacturing industry; financial performance; corporate finance; corporate innovation; financial liquidity; Fourth Industrial Revolution; Structural Equation Modeling

## 1. Introduction

This study aims to examine the mediation channels through which the financial performance of intelligent manufacturing industries closely related to the Fourth Industrial Revolution has been affected.

Along with the emerging Fourth Industrial Revolution which was made clear in the Davos Forum in 2016, several countries in the world started to proactively respond to the revolution. Industry 4.0 by Germany, Advanced Manufacturing Partnership (AMP) by the USA, and Made in China 2025 by China are the well-known plans each of the governments have put forward. The plans mainly focus on intelligent manufacturing, which has been made possible by the key technologies of the Fourth Industrial Revolution, such as big data, Internet of Things (IoT), and artificial intelligence, and the advances in the connectivity of the technologies.

According to the Manufacturing Trends Report (2019) issued by Microsoft, the global intelligent manufacturing market size was 59.15 billion dollars in 2018, and the projected global market will arrive at 74.80 billion dollars by 2022. China also is supposed to show great development in intelligent manufacturing. As of April 2019, Statista published on its website that China's intelligent manufacturing market size was 33.75 billion dollars (2250 billion yuan) in 2019 and would amount to approximately 58.02 billion dollars (3868 billion yuan) in 2022. All these changes have led enterprises and governments to

increase their attention to the investments and innovations in the intelligent manufacturing industry.

As of 17 April 2020, the Ministry of Science and Technology of China (2020) announced the national enterprise innovation survey results on its official website. According to the survey report, the number of enterprises carrying out innovation activities nationwide accounted for 40.8% of all enterprises in 2018. The enterprises of the manufacturing industry in particular have not only the highest degree of innovation activity, but also the highest success rate of innovation activities. Moreover, large-scale high-tech enterprises, which are usually associated with intelligent manufacturing, have outstanding innovation capabilities and play a leading role in manufacturing.

The transforming of the production system into intelligent manufacturing, which is supposed to require more innovation, is alleged to give enterprises better chances to develop in the long run, while could give rise to a financial burden for them in the process and eventually affect their financial performance, i.e., the final goal of enterprises. Nonetheless, there has not yet been much research on that subject regarding intelligent manufacturing all over the world, and much less in China, as the Fourth Industrial Revolution is in the early stage of development globally.

Although we recently saw an increase in the research on intelligent manufacturing industries or Industry 4.0, there has not been much focus on the performance of intelligent manufacturing, especially for China. Jie et al. (2020), recent rare research on that subject, deals with the intelligent manufacturing industry of China; it focuses on the impact of intelligent manufacturing itself on financial performance and innovation performance, separately. However, there is not yet research on the causal relationship between a few main influencing factors including innovation and financial performance in the intelligent manufacturing industries, nor on the mediation channels that affect financial performance.

Therefore, it is crucial to study the innovation of China's intelligent manufacturing enterprises and the mediation channels through which their performance, especially the financial performance, is affected. This study seeks to determine whether innovation, the most commonly mentioned factor for improving enterprise performance in the intelligent manufacturing industry, alone contributes sufficiently to enhancing financial performance.

There are more than a few factors affecting the financial performance of enterprises, so diverse is the influence of the factors on business performance. This study aims to establish a theoretical model concerning the impacts of external factors such as the economy, politics, and regional innovation capability on enterprises' internal innovation, mainly led by R&D investment, and the impacts of the internal innovation on the enterprises' liquidity and financial performance as a whole.

The external environment, also called the macro environment of industry, refers to economic, political, social, regional, and innovative factors that affect the industry and enterprises. According to the article published on the Twproject blog, the external environment is composed of factors that occur outside the enterprise but which can cause internal changes and are beyond the company's control. Even the external environment occurs outside an enterprise; it can have a significant influence on the enterprise's current operations, growth, and long-term sustainability. (Blog article in https://twproject.com/blog/internal-external-corporate-environmental-factors-project-environment/, accessed on 3 July 2020).

As an external element, the economy is one of the most important factors for the success of an enterprise. Some contributing factors such as the fluctuation of the interest rate and economic crisis directly and strongly affect the consumption and consequently the profits of businesses. Government financial supports (usually in the form of subsidies), as an essential measurement for the government to implement macro-control in China, can also increase the cash inflow of enterprises and thus enhance their solvency (Wang 2017).

An excellent regional innovation environment can promote a more effective combination of innovation system elements and enhance the efficiency of scientific and technological innovation. For instance, Francesco Quatraro (2009) investigated the relationship between

the diffusion speed of the regional innovation capabilities and the start period of industrialization in Italian regions, and the research results show that the diffusion of the innovation capabilities in late-industrialized regions is faster than that in the early-industrialized ones.

R&D and technological innovation also play essential roles in the development of enterprises, the economy, and the nation, ranging from product improvement, process productivity improvement, and new market development for new businesses (Lee et al. 2005). In addition, for an enterprise, liquidity is one of the most important goals of working capital management and the central task of revenue optimization and financial performance. Eljelly (2004) argued that liquidity management is essential when an enterprise is in excellent condition but is most significant during the period enterprise got troubles.

Therefore, this paper integrates the main external and internal factors of the enterprise from the above perspectives to analyze their mediation channels on enterprise performance.

The more specific research purposes are as follows.

The first is to examine the impact of China's economic development, infrastructure, human resources, and related government policies on the technological research of enterprises related to the Fourth Industrial Revolution.

The second is to analyze the impact of the enterprises' ability to develop intelligent new technology and the invested human and financial resources on the enterprises' overall financial performance.

The third is to confirm the mediating effect of the enterprises' internal R&D investment on the overall financial performance by setting the enterprises' liquidity (from the perspective of financial information) as a mediator.

Finally, through statistical analysis, by applying the data obtained in the survey, this study proposes the data's academic and practical significance and implications, particularly the situations and aspects related to the era of the Fourth Industrial Revolution. Under these considerations, feasible references and recommendations are suggested on how to optimize the allocation of resources, and how to adjust the internal operating structure and financial strength to enhance the capability of enterprises and to improve their financial performance.

For this study, we select the 317 enterprises which are related to intelligent manufacturing and listed in the China Stock Market as research objects.

To accomplish the research purposes, this study conducts both a literature review and a field survey for statistical analysis. The empirical analysis of this study mainly uses three kinds of statistical software. First, we use the SPSS20.0 for frequency analysis to examine the general characteristics of enterprises that responded to the field survey. Then, we use the R Program to conduct Exploratory Factor Analysis (EFA), Confirmatory Factor Analysis (CFA), Correlation Analysis, and verification of the structural model to verify the hypotheses in the research model. We adopt both the R Program and SmartPLS to double-check the reliability and validity of the variables, especially in the verification of the reliability and validity constructs. Finally, we use the LISREL program to further explain the mediating effects of variables.

## 2. Theoretical Background

### 2.1. The Fourth Industrial Revolution and China's Intelligent Manufacturing

Davis (2016) pointed out that the entire world is in the early stages of the Fourth Industrial Revolution. According to Michael (2019) on Geospatial World's blog, the characteristics of the Fourth Industrial Revolution are different in speed, scope, and influence compared with the existing First to Third Industrial Revolutions. The article James Patterson (2020) wrote on the blog Transcosmos reveals that technology will advance at breakneck speed, lead to significant reorganization of all industries and bring about significant change through the systems of society, including production systems and governance structures.

Advanced intelligent technologies related to the Fourth Industrial Revolution appear in various ways, so diverse are the fields of IT technology innovation and production technology innovation. Based on a survey of chief human resources officers and top strategy

executives from the companies across 12 industries and 20 developed and emerging economies (which collectively account for 70 percent of global GDP), the World Economic Forum (2018) issued the report named "Future of Jobs Report 2018". The report shows the proportions of enterprises in technologies that are projected to be adopted by 2022, where the top is Big Data, followed by Internet of Things, Machine Learning (Artificial Intelligence), Virtual Reality, Blockchain, 3D Printing, Robotics, and others. Therefore, considering the significance and representativeness of the report, this study selects the enterprises that mainly applied these technologies as the survey objects.

As a developing country, China has actively introduced strategies in response to the arrival of the Fourth Industrial Revolution and shortened the gap with other developed countries. In March 2015, Li Keqiang, the current premier of China, first suggested the plan "Made in China 2025" in the "Report on Government Work"; accordingly, the government document "Made in China 2025" was officially released in the same year.

Xie et al. (2020) points out that the Fourth Industrial Revolution, also known as Industry 4.0, leads even to the formation of the intelligent supply chain. Chen et al. (2020) also suggests that for the small and medium-sized enterprises in China, intelligent manufacturing involves intelligent equipment, production optimization, intelligent service automation, and industrial information integration, which need diverse technologies such as IoT, cloud computing, and big data.

According to the World Intelligent Manufacturing Center Development Trend Report (2019), as of 2019 China had a total of 537 intelligent manufacturing industrial parks, covering 27 provinces and cities across the country. Moreover, according to the China Intelligent Manufacturing Industry Development Report (2019), as of 2019 there were more than 9000 intelligent manufacturing enterprises in China. From a geographical perspective, China's intelligent manufacturing has been concentrated in four major regions: the Bohai Rim Region, the Yangtze River Delta Region, the Pearl River Delta Region and the Midlands and Southwestern Region. In China, three major economic circles have been formed, including the Bohai Rim Economic Circle (BREC), the Yangtze River Delta, and the Pearl River Delta. Each of these circles is centric to one of China's megacities: Beijing (BREC), Shanghai (Yangtze River Delta), and Guangzhou (Pearl River Delta) (Wen et al. 2018). The Midlands and Southwestern Region is a recent emerging economic circle, each of which has its own characteristics.

### 2.2. Literature Review

Hofer (1983) has argued that financial performance indicators are objective indicators and are the most common performance indicators in strategic research, since they are used in various fields to study management performance. The enterprise financial performance indicators are objective indicators that can be used as general indicators, such as sales growth rate, net profit growth rate, return on sales, and return on investment (Pearce et al. 1987; Shrader et al. 1984). This study grasps the research related to enterprise financial performance from three aspects: environmental factors such as economic, political, and technological factors outside enterprise, enterprise R&D investment, and financial liquidity of enterprises.

In previous research on the environmental factors affecting enterprise performance, Daft et al. (1988) have proposed technology, policy, and sociocultural environment. Yi and Liu (1999) have also proposed environmental factors, such as administrative law, social culture, globalization trends, and technology. Also, Johns and Cena (2018) has pointed out that there are two types of external environments: the macroenvironment of economy, society, culture, politics, and technology, and the microenvironment of suppliers, customers, and public perception.

For high-tech enterprises, government subsidies can encourage them to increase R&D investment and then improve the overall business performance (Zhang et al. 2016). Besides, Zhang et al. (2016) pointed out that the financial support of the Chinese government has continued to grow in recent years. However, some empirical studies have shown

that government subsidies are ineffective and even play a negative role in the innovative activities of enterprises and their performance (Xu et al. 2014)

An excellent regional innovation environment can provide various external conditions required by innovation entities, promote a more effective combination of innovation system elements and enhance the efficiency of scientific and technological innovation. Marshall (1890) first argued that the regional industrial agglomeration is conducive to the transfer and spread of new knowledge and new information between upstream and downstream enterprises. Jacobs (1969) also has argued that the variety of industries within a geographic region promotes knowledge externalities and ultimately innovative activity and economic growth. In China, Liu and Chai (2011), Cheng and Liu (2015), and Li et al. (2016) also empirically analyzed the influence of different industries' agglomeration on enterprises' technological innovation capabilities and proved that there is a significant correlation between the two factors.

Technological innovation is also increasingly vital for the survival and development of enterprises, and the innovation and learning capability is also very crucial in supply chain performance (Xie et al. 2020). R&D and technological innovation play an essential role in the development of enterprises, the economy, and the nation (Lee et al. 2005). However, Venkatraman and Prescott (1990) have argued that there is no statistically significant relationship between the size of an entity's R&D investment and its financial performance. McCutchen and Swamidass (1996) have also pointed out that there is no significant positive relationship between the size of an enterprise's R&D investment and its market value. Jaruzelski et al. (2005) also have argued that, for Apple Group, its R&D intensity was 5.9% in 2004, lower than the industry average of 7.6%, but its high enterprise performance was achieved by continuing to release innovative products through a selective concentration of resources.

From the aspect of enterprise financial liquidity, liquidity is one of the most important goals of working capital management. Efficient working capital management helps to improve the operating performance of business concerns and satisfy short-term liquidity (Maness and Zietlow 2005; Samiloglu and Demirgunes 2008).

Enekwe et al. (2014) have examined the relationship between financial leverage and enterprise financial performance in Nigeria, which shows that the independent variables, including Debt Ratio, Debt-Equity Ratio, and Interest Coverage Ratio, have no significant impact on financial performance. The study conducted by Titman and Wessels (1988) is on the determinants of the capital structures in the US-based manufacturers within the SEM framework, which shows the significant explanatory power of profitability and enterprise size concerning an enterprise's capital structure.

In China, Xu and Xian (2002) have examined the effect of bankruptcy on the market behavior of enterprises, where the authors have found that enterprises with more intangible assets would carry out more debt financing. Also, Crutchley and Hansen (1989) have tested the financial qualities of 603 companies by adopting the Ordinary Least Squares (OLS) method to identify how financial indicators affect the company's capital structures. The study results suggested that both the percentage of R&D expense and the company size have a significant impact on the company's capital structure.

## 3. Research Model and Analysis Methods

### 3.1. Methodology of SEM

This study uses SEM, which is a multivariate statistical technique, to analyze structural relationships, as an analytical tool. This technique is a combination of factor analysis and multiple regression analysis and is used to examine the structural relationship between measured variables and latent constructs (Gao 2006). In other words, an SEM is a system of linear equations among several unobservable variables (constructs) and observed variables (Sinharay 2010), which are also called endogenous variables and exogenous variables.

As shown in Figure 1, shown by Sartal et al. (2017), there are two types of models: a measurement model and a structural model. A measurement model defines latent

variables (F1, F2) using one or more observed variables (X1, X2, X3; Y1, Y2, Y3), and a structural model imputes relationships between latent variables. The measurement model represents the theory that specifies how measured variables come together to represent the theory. The measurement model of SEM allows the researcher to evaluate how well the observed (measured) variables of the study combine to identify the underlying hypothesized constructs. Confirmatory Factor Analysis is applied to test the measurement model, and the hypothesized factors are also called latent variables (Weston and Gore 2006). The structural model represents the theory that shows how the constructs are related to the other constructs. Equations in the structural portion of the research model specify the hypothesized relationships among latent variables.

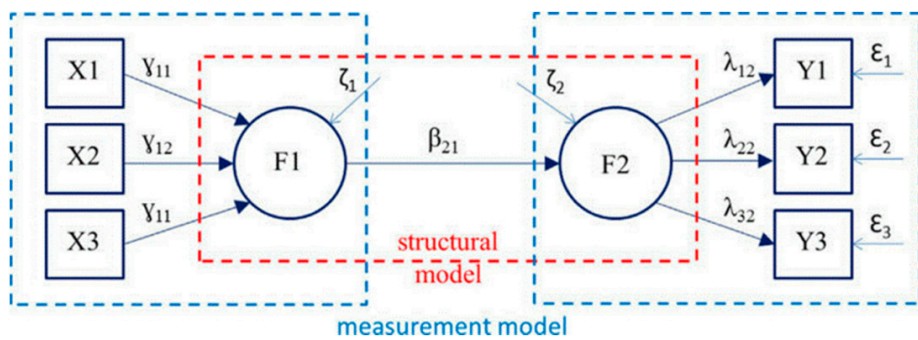

**Figure 1.** Measurement model and structural model in SEM.

A dissertation editing service institution, Statistics Statistics Solutions (2020), listed that there are generally four steps in SEM analysis: first, developing the overall measurement model, which is also known as path analysis; second, designing the study to produce empirical results by specifying the structural model; third, conducting the CFA to assess the measurement model validity; and fourth, examining the structural model validity by confirming the test results under a series of goodness-of-fit indices (such as the Adjusted Goodness of Fit Index (AGFI), Comparative Fit Index (CFI), and Goodness of Fit Index (GFI)) and badness-of-fit indices (such as the Root Mean Square Residual (RMR), Root Mean Square Error of Approximation (RMSEA), and Standardized Root Mean Square Residual (SRMR)) that meet the predetermined criteria.

Gao (2006) believes that SEM has advantages compared with the traditional statistical modelling analysis methods, and the independent variables of the regression equation allowed measurement errors to be contained and multiple dependent variables to be processed. Moreover, the measurement of factors and the structure between factors can be dealt with simultaneously in one model. This kind of model can not only test the reliability and validity of the factors, but also integrate the concept of measurement reliability into the statistical inferences, such as the path analysis.

### 3.2. Research Model and Hypotheses

Based on the theoretical literature review and the basic principles, structure, and characteristics of the SEM model, this study designs, as Figure 2 shows, the conceptual framework of each variable and constructs the following research model for empirical analysis and parameter estimation. The corresponding six research hypotheses of this paper are summarized in Table 1.

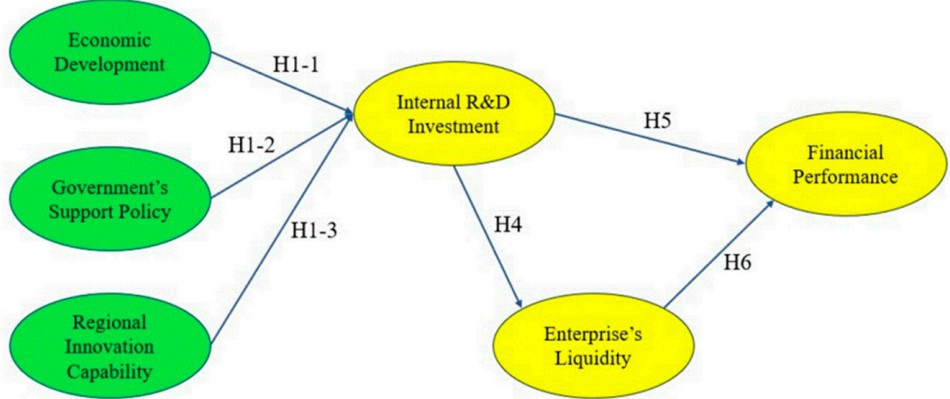

**Figure 2.** Proposed theoretical research model.

**Table 1.** The summary of research hypotheses.

| No. | | Hypotheses |
|---|---|---|
| H1 | H1-1 | Urban economic development has a significant and positive impact on an enterprise's internal R&D investment. |
| | H1-2 | Government's support policy has a significant and positive impact on an enterprise's internal R&D investment. |
| | H1-3 | Regional innovation capability has a significant and positive impact on an enterprise's internal R&D investment. |
| H4 | | There is a significant positive relationship between an enterprise's internal R&D investment and the enterprise's liquidity. |
| H5 | | There is a significant positive relationship between an enterprise's internal R&D investment and the enterprise's financial performance. |
| H6 | | There is a significant positive relationship between an enterprise's liquidity and the enterprise's financial performance. |

The corresponding research hypotheses are as follows.

First, the factors that affect the development and use of new intelligent technologies in the context of the Fourth Industrial Revolution are mainly focused on the external or environmental factors outside of an enterprise. Both Yang (2019) and Liu et al. (2019) believe that the excellent economic development environment and the regional innovation environment have an essential impact on the development of an enterprise's scientific and technological innovation. Also, Xu (2018) tested the inductive effect of the government's R&D investment on the enterprises' R&D investment. Therefore, this study puts forward the economic development, government's support policies, and technological innovation capability of the designated regions as the three exogenous variables in the SEM. Accordingly, three hypotheses, H1-1, H1-2, and H1-3 are proposed for testing in this paper.

Next, Wang et al. (2007) have argued that the asset allocation structure and working capital management efficiency significantly affect the enterprise liquidity. Chan et al. (2003) and Li et al. (2014), among others, have argued that R&D investment brings significant uncertainty to enterprises. Accordingly, this paper proposes the H4 hypothesis to analyze the impact channel of an enterprise's internal R&D investment and liquidity.

Then, Hanel and St-Pierre (2002) found that R&D investment has a direct impact on enterprise efficiency under the condition that intellectual property rights are adequately protected. Zhu and Huang (2012) pointed out that R&D investment has a positive relationship with the return on assets (ROA); that is, R&D investment has a positive impact on enterprise profitability. Taking into consideration these research results and the new industrial model, such as the emergence and development of intelligent manufacturing, further confirmation and analysis would be needed to confirm whether such research

conclusions are still valid. Therefore, this paper suggests the H5 hypothesis to explore the relationship between R&D investment and financial performance of enterprises.

Last, Bao (2015) has argued that in financially constrained enterprises, the sensitivity of cash holdings to cash flow will increase sharply and the structure of an enterprise will have an impact on the enterprise's cash holdings. Yang et al. (2009) have found that the liquidity of an enterprise has a positive impact on the overall operating income. Therefore, this paper puts forward the H6 hypothesis based on this theory.

In addition, to explore the mediating effects and to confirm empirically the comprehensive interaction relationships, this study sets the enterprise's internal R&D investment and enterprise's liquidity as mediator variables.

### 3.3. Data Collection and Survey Overview

To verify the research model and to test the hypotheses, this study establishes a survey project for statistical analysis. Table 2 shows the composition of the survey items. The survey is grouped by each factor and 15 items are arranged for the two variables and 5 items for the general characteristics of the samples. The variables are measured by amount and ratio, and general characteristics are nominal ordinal.

**Table 2.** Composition of survey items by variable types.

| Types | Variables | Items |
|---|---|---|
| Exogenous Variables | Economic Development | City's GRDP |
| | | Urbanization Rate |
| | Government's Support Policy | Government Subsidies for Enterprise R&D |
| | | Financial Support for Science Park Constructions |
| | Regional Innovation Capability | Human and Resource Environment Index of Regional Innovation |
| | | Institutional Service Index of Regional Innovation |
| Endogenous Variables | Internal R&D Investment | Ratio of R&D Investment to Operating Income |
| | | High-level Scientific Research Personnel |
| | | University-enterprise Cooperation Projects for Technological Innovation |
| | Enterprise's Liquidity | Current Ratio (Working Capital Ratio) |
| | | Cash Ratio |
| | | Owner's Equity Ratio |
| | Financial Performance | Profit Margin Ratio |
| | | Return on Assets |
| | | Social Contribution Rate |
| General Characteristics | | Attributes |
| | | Industry Classifications |
| | | Location by Economic Circles |
| | | Applied Technologies |

When applying the SEM, the sample size will have a statistically significant impact on the final analysis results. Barrett (2007) suggests that articles with a sample size above 200 can be accepted, except in cases where the study has a strict quantitative limit. Jackson (2003) believes that with the application of the N:q rule, it can roughly determine the number of samples required: N is the number of samples, and q is the parameter that needs to be estimated in the model. The recommended ratio is 20:1 and can also relax to 10:1

to ensure the estimates of the parameters are credible and guarantee the validity of the significance test. Therefore, this study determines 300 as the appropriate sample size.

To guarantee the credibility and significance of the data analysis results, this study selects samples for statistical analysis using the following approach. In 2019, the China Enterprise Confederation/China Enterprise Directors Association released a ranking of the top 500 Chinese manufacturing enterprises. The evaluation indicators mainly include enterprise size, operating income, market share, and innovation capacity. In other words, these manufacturing enterprises can best represent the most advanced and highly intelligent in all in China. Therefore, the 317 of these enterprises selected for analysis are sufficiently qualified to represent the intelligent manufacturing industry in China.

Since this study explores the influencing channels between different factors and time is not considered one of the factors, this study applies cross-sectional research design; in other words, we collect cross-sectional data for each survey item. As the statistical data used in this research is quantitative, all the data is required to be highly accurate and reliable. Therefore, the data has been compiled mainly through the following prudent approaches.

Above all, this paper collects the relevant data published by the authoritative institutions of China in 2019. These specifically include the statistical yearbook issued by the National Bureau of Statistics of China and China's High-tech Industry Institute, the statistical data released by the China Industrial Information Research Institute, the information published by China's Business Information Network and the Economic Statistics Bulletin of China's provinces.

Then, as the research objects of this paper are listed companies in China, this paper selects the accounting and financial data publicized by China Stock Exchanges, the Industrial Bank of China and other relevant institutions and organizations. Also, this research collects data from the published quarterly and the audited annual financial reports of enterprises.

Last, this paper compares the data published by the institutions and the organizations with the data released by the enterprises to further confirm the consistency of the data.

## 4. Results of Empirical Analysis

### 4.1. General Characteristics of the Samples

According to the announcement on China Finance Net (Founded in 2002, China Finance Net is an online news and information center in China's financial sector, and now is China's leading provider of financial data and financial information. Find more details of the announcement on http://www.financeun.com/newsDetail/28806.shtml?platForm=jrw, accessed on 15 July 2020), at the 2019 Shenzhen Stock Exchange Technology Conference, the China Securities Regulatory Commission (CSRC) announced that nearly half of the listed companies in China's Shanghai and Shenzhen stock markets are high-tech companies, and the high-tech industry in the Second-board Market accounts for more than 90%. Moreover, in China, intelligent manufacturing and other high-tech industries are closely related to the Fourth Industrial Revolution that the Chinese government vigorously promoted and developed. In other words, studying the listed enterprises in China's high-tech industries has more important research significance and is more representative. Therefore, we select the listed enterprises as the survey objects.

This research conducted a preliminary survey from March 2019 and the full-scale poll was conducted both online and offline in China from April to June 2019. This study eventually adopts 317 listed enterprises as the statistical analysis samples. As Table 3 shows, the general statistical characteristics of the samples are as follows.

**Table 3.** General characteristics of the samples.

| Categories | | Frequency (N = 317) | Percentage (%) |
|---|---|---|---|
| Attributes | Listed Enterprise | 317 | 100 |
| | Unlisted Enterprise | 0 | 0 |
| Industry Classifications | Machinery Manufacturing | 143 | 45.11 |
| | Information and Communications Technology | 92 | 29.02 |
| | Biomedicine and Instruments | 20 | 6.31 |
| | New Material | 34 | 10.73 |
| | Transportation and Logistics | 20 | 6.31 |
| | Others | 8 | 2.52 |
| Locations by Economic Circles | Bohai Rim Economic Circle | 60 | 18.93 |
| | Midlands Economic Circle | 45 | 14.20 |
| | Yangtze River Delta Economic Circle | 116 | 36.60 |
| | Pearl River Delta Economic Circle | 78 | 24.60 |
| | Southwest Economic Circle | 13 | 4.10 |
| | Others | 5 | 1.58 |
| Applied Technologies | Internet of Things | 214 | 25.91 |
| | Artificial Intelligence | 152 | 18.40 |
| | 3D Printing | 50 | 6.05 |
| | Big Data | 310 | 37.53 |
| | Robotics | 100 | 12.10 |
| Post-listing Operation Period | 1~5 years | 32 | 10.07 |
| | 5~10 years | 217 | 68.50 |
| | over 10 years | 68 | 21.43 |

First, by the industry classifications, machinery manufacturing accounts for the highest proportion, reaching 45.11%. The second is information and communications technology, accounting for 29.02%, followed by the industry of New Material (10.73%), Biomedicine and Instruments (6.31%) and Transportation and Logistics (6.31%).

Second, by the locations of the enterprises by the economic circles, there is no significant gap in the proportion of the economic circles, except for the southwest economic circle (4.10%). In comparison, the Yangtze River Delta Economic Circle accounts for the most significant proportion of 36.60%, followed by the Pearl River Delta Economic Circle (24.60%), the Bohai Economic Circle (18.93%), and the Midlands Economic Circle (14.20%).

Third, of the technologies employed by the enterprises, the most widely developed and applied are "Big Data" and "Internet of Things", which account for 37.53% and 25.91%, respectively. Next, the proportion of Artificial Intelligence accounts for 18.4%, then Robotics and 3D Printing technology account for 12.10% and 6.05%, respectively.

Fourth, more than half (68.50%) of the enterprises have been listed on the stock exchange market for five to ten years. Meanwhile, enterprises with more than ten years account for 21.43%, and only 10.07% of the enterprises have been listed for less than five years.

### 4.2. Analysis of Measurement Model

4.2.1. Reliability Analysis

A scientific and valid questionnaire serves as the basis for the investigation and analysis of statistics, and so a higher reliability coefficient stands for more consistent, stable and reliable test results. This paper has ensured sampling adequacy, reliability of the data and the accuracy of the research results through multiple indicators. As Table 4 shows, the results of the analysis are as follows.

**Table 4.** Test results of Reliability Analysis.

| Variables | Observed Variables | Factor Loading | Cronbach's Alpha | rho_A (ρA) |
|---|---|---|---|---|
| Economic Development | Eco1<br>Eco2 | 0.90<br>0.96 | 0.969 | 0.971 |
| Government's Support Policy | Gov1<br>Gov2 | 0.78<br>0.97 | 0.877 | 0.931 |
| Regional Innovation Capability | Tech1<br>Tech2 | 0.96<br>0.71 | 0.858 | 0.930 |
| Internal R&D Investment | Rnd1<br>Rnd2<br>Rnd3 | 0.84<br>0.76<br>0.78 | 0.801 | 0.814 |
| Enterprise's Liquidity | Liqui1<br>Liqui2 | 0.95<br>0.82 | 0.850 | 0.894 |
| Financial Performance | Perf1<br>Perf2<br>Perf3 | 0.71<br>0.80<br>0.81 | 0.820 | 0.946 |

First, this study verifies the goodness-of-fit through the Kaiser-Meyer-Olkin (KMO) sample fit and Bartlett's (1951) Test for Sphericity. The KMO value of all the variables in this study turns out to be 0.849, which appears a reasonable level. Therefore, this study judges that the variables selected for factor analysis are acceptable and the reliability is high. The verification for the Bartlett (1951) test also is suitable with a significance level of 0.000.

Second, in the first EFA analysis, item 3 (Liqui3) in the factor named enterprise's liquidity is removed because the value of factor loading is lower than the baseline (0.45). Secondary analysis results show that the validity of all the variables is confirmed, as all the values of factor loading turn out to be more than 0.70 statistically.

Third, regarding the evaluation of the measurement model, this study applied both the Cronbach's Alpha coefficient and the Dijkstra-Henseler rho_A (ρA) as reliability measurement indicators. According to DeVellis (1991) and Nunnally and Bernstein (1994), the rule of thumb for test results of Cronbach's Alpha is that 0.70 or higher is the recommended criterion. As Dijkstra and Henseler (2015) state, Cronbach's Alpha is not consistent, and researchers should evaluate and report construct reliability by means of the Dijkstra-Henseler's rho_A (ρA) reliability coefficient (Dijkstra and Henseler 2015). Similar to Cronbach's Alpha, the reliability coefficient ρA should indicate values of 0.70 or higher in exploratory research and values above 0.80 or 0.90 for more advanced stages of research (Hair et al. 2011; Henseler et al. 2009; Nunnally and Bernstein 1994). As shown in Table 4, the reliability coefficient Cronbach's Alpha and ρA of each measurement construct is all above 0.80, which means the reliability of this study is relatively excellent.

### 4.2.2. Validity Analysis

Analysis of reliability ensures the reliability of sample data, while the validity analysis guarantees the accuracy and validity of the model and determines the accuracy of questionnaire items. Validity Analysis is mainly confirmed by CR, AVE, and factor loading of variables. Generally, only when the CR value is above 0.7, the AVE value is above 0.5 and all the standardized factor loadings of variables are above 0.7, can researchers rate the convergent validity as good (Woo 2012). CFA is a statistical tool that allows for the assessment of the fit between the observed data and a priori conceptualized, the theoretically grounded model that specifies the hypothesized causal relations between latent factors, and their observed indicator variables (Hancock and Mueller 2001). Therefore, this study conducts CFA, and the results are shown in Table 5.

**Table 5.** Test results of Validity Analysis.

| Latent Variables | Items | Factor Loading | Std.Err | z-Value | SMC | Std.all | p-Value | CR | AVE |
|---|---|---|---|---|---|---|---|---|---|
| Economic Development | Eco1 | 1.000 | - | - | 0.850 | 0.922 | - | 0.975 | 0.950 |
| | Eco2 | 1.165 | 0.040 | 29.139 | 0.787 | 0.870 | 0.000 | | |
| Government's Support Policy | Gov1 | 1.000 | - | - | 0.692 | 0.701 | - | 0.941 | 0.895 |
| | Gov2 | 1.708 | 0.267 | 13.407 | 0.769 | 0.812 | 0.000 | | |
| Regional Innovation Capability | Tech1 | 1.000 | - | - | 0.627 | 0.726 | - | 0.956 | 0.841 |
| | Tech2 | 1.045 | 0.195 | 26.521 | 0.725 | 0.869 | 0.000 | | |
| Internal R&D Investment | Rnd1 | 1.000 | - | - | 0.714 | 0.905 | - | 0.827 | 0.674 |
| | Rnd2 | 0.885 | 0.070 | 7.624 | 0.619 | 0.746 | 0.000 | | |
| | Rnd3 | 0.914 | 0.057 | 6.308 | 0.703 | 0.724 | 0.000 | | |
| Enterprise's Liquidity | Liqui1 | 1.000 | - | - | 0.803 | 0.896 | - | 0.873 | 0.675 |
| | Liqui2 | 0.806 | 0.095 | 10.824 | 0.711 | 0.760 | 0.000 | | |
| Financial Performance | Perf1 | 1.000 | - | - | 0.695 | 0.761 | - | 0.898 | 0.723 |
| | Perf2 | 0.762 | 0.058 | 11.858 | 0.701 | 0.713 | 0.000 | | |
| | Perf3 | 0.946 | 0.079 | 29.949 | 0.837 | 0.874 | 0.000 | | |

The verification results show that both the composite reliability and all the standardized factor loadings (Std.all) are above the standard reference values, indicating that there is internal consistency. Meanwhile, the AVE values are also higher than the standard reference values, indicating that there is no problem in reliability statistically. Therefore, the observed variables in this study describe and reflect the latent variables well. The other items also show good results; for instance, ethical SMC values for latent variables and the z values are all significant since all values are above 1.965.

*4.3. Analysis of Structural Model*

4.3.1. Structural Model Fit

This study analyses the value of the path coefficient, which can confirm the goodness-of-fit of the structural model based on the CFA to determine the statistical superiority of the structural model. Bentler and Chou (1987), Kline (2015) and others distinguish between several types of fit indices: absolute fit index, incremental fit index, parsimony fit indices, and those based on the non-central parameter. Bollen (1990) explains both absolute fit index and incremental fit index, such as the Bentler-Bonett Index, or NFI, CFI, RMSEA and SRMR. Therefore, this study selects a total of eight indices from the absolute fit index and the incremental fit index to judge the fitness of the structural model. Table 6 shows the final analysis results.

**Table 6.** Confirmation of structural model fit.

| Goodness of Fit Index | | | Structural Model | Confirmation Criteria |
|---|---|---|---|---|
| Absolute Fit Measures | Overall Fitness of the Model | $x^2/df$ | 1.893 | $1.0 \leq x^2/df \leq 2.0{\sim}3.0$ |
| | | RMSEA | 0.067 | $\leq 0.05{\sim}0.08$ |
| | | RMR | 0.045 | $\leq 0.05{\sim}0.08$ |
| | Explanatory Power of the Model | GFI | 0.914 | $\geq 0.90$ |
| | | AGFI | 0.867 | $\geq 0.80{\sim}0.90$ |
| Incremental Fit Measures | | NFI | 0.908 | $\geq 0.90$ |
| | | CFI | 0.939 | $\geq 0.90$ |
| | | IFI | 0.940 | $\geq 0.90$ |

As the measured absolute fit index, $x^2/df$ = 1.893, RMR = 0.045, RMSEA $\chi$ = 0.067, GFI = 0914, and AGFI = 0.867; the goodness-of-fit indices all meet the threshold criteria.

As incremental fit indices, NFI = 0.908, CFI = 0.939, and IFI = 0.939 also indicate that the goodness-of-fit indices all meet the threshold criteria. Therefore, the result of the path analysis of this study is reliable, which means that the causal relationship between variables is consistent.

### 4.3.2. Verification of Hypotheses

This study puts forward six factors in the path analysis to systematically understand the causal relationship between the factors. To verify the hypotheses, this study verifies the SEM using the Maximum Likelihood Method of the R program, and Figure 3 shows the final results of path analysis.

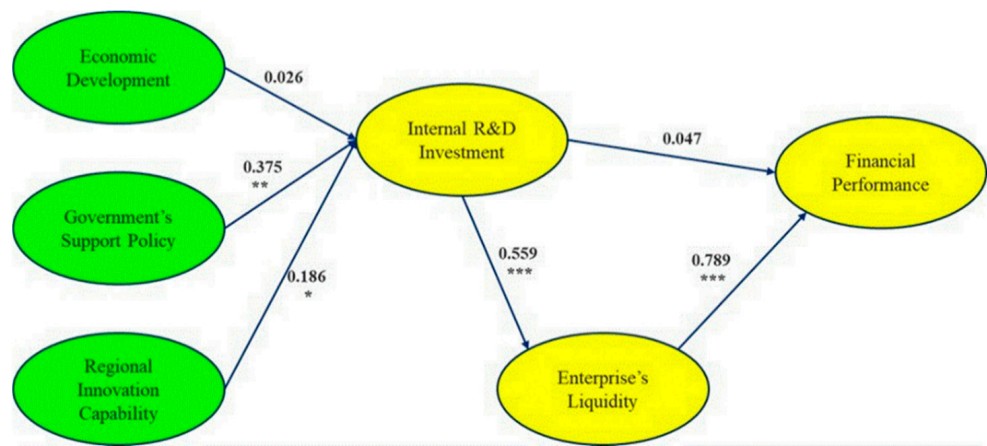

**Figure 3.** The final results of path analysis. Symbol Meaning: symbols *, **, *** indicating statistical significance. When there is non-symbol means *p* value more than 0.05 shown as "*p* > 0.05"; symbol * means $p \leq 0.05$; ** means $p \leq 0.01$; *** means $p \leq 0.001$.

Here are the specific results of the statistical verification in Table 7, namely, Standardized Estimate, S.E., *z*-value and *p*-value. From the overall results, four of the six hypotheses from H1-1 to H6 established in this study are accepted, while two are rejected, statistically; the accepted hypotheses are H1-2, H1-3, H4, and H6, and the rejected are H1-1 and H5.

**Table 7.** The results of SEM analysis and hypotheses verification.

| Hypotheses | | | Est. Std | S.E. | *z*-Value | *p*-Value | Results |
|---|---|---|---|---|---|---|---|
| | H1-1 | Economic Development → Internal R&D Investment | 0.026 | 0.075 | 0.342 | 0.732 | Rejected |
| H1 | H1-2 | Government's Support Policy → Internal R&D Investment | 0.375 | 0.061 | 4.641 | 0.009 | Accepted |
| | H1-3 | Regional Innovation Capability → Internal R&D Investment | 0.186 | 0.064 | 2.509 | 0.012 | Accepted |
| H4 | | Internal R&D Investment → Enterprise's Liquidity | 0.559 | 0.077 | 10.846 | 0.000 | Accepted |
| H5 | | Internal R&D Investment → Financial Performance | −0.047 | 0.084 | −0.840 | 0.625 | Rejected |
| H6 | | Enterprise's Liquidity → Financial Performance | 0.789 | 0.090 | 18.416 | 0.000 | Accepted |

In H1-1, the standardized regression coefficient (Est.Std) of the economic development to the internal R&D investment is 0.026, the *z*-value is 0.342, and the *p*-value is 0.732, which does not satisfy the baseline of the significance level. That is, urban economic development has no significant and positive impact on an enterprise's internal R&D investment.

In H1-2, the Est. Std is 0.375, the *z*-value is 4.641, and the *p*-value is 0.009, which satisfies the baseline of the significance level. It means the government's support policy has a significant and positive impact on an enterprise's internal R&D investment.

In H1-3, the Est. Std is 0.186, the *z*-value is 2.509, and the *p*-value is 0.012, which reaches the significance level. It means the regional innovation capability has a significant and positive impact on an enterprise's internal R&D investment.

In H4, the Est. Std is 0.559, *z*-value is 10.846 and the *p*-value is 0.000, which means there is a significant positive relationship between an enterprise's internal R&D investment and the enterprise's liquidity.

In H5, the Est. Std of the internal R&D investment to the financial performance is −0.047, the *z*-value is −0.840, and the *p*-value is 0.625, which does not reach the significance level. It indicates that there is no significant positive relationship between the two factors statistically.

In H6, the Est. Std of an enterprise's liquidity to the financial performance is 0.789, *z*-value is 18.416, and the *p*-value is 0.000, which means enterprise liquidity has a significant and positive impact on financial performance.

### 4.4. Confirmation of Mediating Effects

A mediating effect occurs when the effect of one variable on a second is mediated, in whole or in part, by one or more other intervening variables (Brown 2006). According to MacKinnon et al. (2010), mediation is a hypothesized causal chain in which one variable affects a second variable that, in turn, affects a third variable.

This analysis mainly confirms the mediating effects from three aspects through seven impact paths. First is the mediation analysis of the external environmental factors on an enterprise's liquidity through R&D investment. Second is the mediation analysis of the external environmental factors on financial performance through R&D investment. Third is the mediation analysis of the R&D investment on financial performance through an enterprise's liquidity. The results of the analysis are presented in Table 8.

**Table 8.** Confirmation of the mediating effects of variables.

| Impact Path | Mediating Effect | Path Coef. | Std. Err | *t*-Value |
|:---:|:---:|:---:|:---:|:---:|
| 1 | ECO → RND → LIQUI | 0.010 | 0.020 | 0.340 |
| 2 | GOV → RND → LIQUI | 0.170 | 0.020 | 2.680 ** |
| 3 | TECH → RND → LIQUI | 0.150 | 0.030 | 2.230 * |
| 4 | ECO → RND → PERF | 0.000 | 0.010 | 0.340 |
| 5 | GOV → RND → PERF | 0.030 | 0.010 | 2.010 * |
| 6 | TECH → RND→ PERF | 0.040 | 0.020 | 1.920 |
| 7 | RND → LIQUI → PERF | 0.280 | 0.060 | 3.910 *** |

$p < 0.05$ *, $p < 0.01$ **, $p < 0.001$ ***. $t > 1.96$ *, $t > 2.58$ **, $t > 3.28$ ***. $p$ values are reported or indicated by *, ** and *** for $p < 0.05$, $p < 0.01$ and $p < 0.001$, respectively. *t*-values represent whether there is statistical correlation. Explain whether the analysis route is established. When *t*-Value is more than 1.96, shown as *, more than 2.58 shown as **, and more than 3.28 shown as ***.

First, in the seventh path, the impact path of the R&D investment on financial performance through enterprise liquidity, the path coefficient is 0.280 and the *t*-value is 3.910, resulting in the most substantial mediating effect ($p < 0.001$). In this paper, working capital ratio, cash ratio, and owner's equity ratio are used as valuation indicators for the variable named enterprise's liquidity. Therefore, this result reveals that the enterprises in intelligent manufacturing industries achieve good financial performance when they combine innovation activities with financial liquidity.

Second, the government's support policy has mediating effects both on an enterprise's liquidity and financial performance through R&D investment, since the *t*-values are 2.680 and 2.010. In particular, the mediating effect on an enterprise's liquidity is statistically more significant ($p < 0.01$). Thus, through R&D activities, the Chinese government's support policy is closely related to the improvement of enterprises' liquidity and performance.

Last, the regional innovation capability has a mediating effect on an enterprise's liquidity through R&D investment with the path coefficient of 0.150 and the *t*-value of 2.230, higher than 1.960. In the corresponding investigation, we find that in the aspect of regional innovation capabilities, China currently takes the promotion policy for the intelligent manufacturing industry as the main development strategy. In this context, high-end manpower and technological infrastructure are particularly important.

By confirming mediating effects, this study highlights a deep understanding of the influencing channels between the environmental factors and an enterprise's internal operation and financial performance. On this account, this study is compelling in seeking better ways to improve an enterprise's technology development and business performance.

## 5. Conclusions

This study focuses on the mediation channels through which the financial performance of intelligent manufacturing industries closely related to the Fourth Industrial Revolution has been affected. In addition to compiling a massive volume of datasets publicized by the Chinese government and other authoritative institutions, a survey on the 317 listed enterprises of the intelligent manufacturing industries in China has been established for statistical analysis. Using the SEM research model, this research tests six hypotheses and confirms the inter-factor impact relationship between exogenous and endogenous variables.

Above all, we find that innovation efforts mainly led by increasing investment in R&D, along with high liquidity, surely lead to good financial performance, whereas innovation efforts alone do not. Considering the high level of R&D investments in the intelligent manufacturing (high-tech) industries analyzed, the study implies that enterprises in the Fourth Industrial Revolution are also required to secure sufficient liquidity in order to achieve successful outcomes from R&D investments. Government support policy has been found to be closely related not only to higher liquidity, but to good financial performance through the common channel of R&D investment, which explains why government's role has been so essential for improving business results in China. Regional innovation capability has been revealed to be related to R&D investments, and furthermore, to liquidity, which shows that the regional innovation system in China has been functioning relatively well to induce enterprises to increase investments and secure higher liquidity, and finally contribute to achieving better business performance. However, regional economic development shows no relationship with R&D investments, and consequently neither with liquidity nor with performance, which infers that innovation activities are not yet considered crucial for development at the regional level in China, though they are at the country level.

Based on the research results, this study reveals a few important implications and puts forward practical suggestions from the following three perspectives: the government, the region, and the enterprise.

From the perspective of the government, the Chinese government needs to increase the financial support for intelligent manufacturing enterprises, which can encourage the enterprises to expand R&D investment further, improve the enterprises' innovation achievements, and ultimately contribute to the overall financial performance. The Chinese government can help the enterprises through additional public investment, as well as by leveraging private investments through government guarantees. Meanwhile, the government can make suitable intervention policies, including the upgrading of school curricula and teacher training, and reinvention of vocational training in the cities. Providing enterprises with business support services, such as workforce training, technology transfer, financial assistance, and sectoral initiatives can also be highly desirable.

From the perspective of the region, it is highly recommended that the regional government enhance the regional innovation capabilities, for they are closely related to the human and resource environment and the institutional service. Capability building in local innovation needs to be given high priority as well. The development of the Fourth Industrial Revolution-related enterprises requires key personnel capable of using new job

skills. On the one hand, it is necessary to introduce the technical expertise and management talents requisite for the development and application of new technologies to cope with the challenges of the rapid improvement of technology and the increasing demands for skills. On the other hand, it is also necessary to improve the knowledge and skills of employees through various training programs to obtain the technical reserves indispensable for intelligent manufacturing.

From the perspective of the enterprise, it is necessary to invest a lot of capital and fixed assets like land, equipment, and property, which have an impact on the capital structure and cash flow of the enterprises, to improve liquidity. Moreover, it helps enterprises to carry out accurate financial analyses and make appropriate economic forecasts so that managers can make more rational decisions. Also, the enterprises can conduct feasibility studies beforehand, strengthen supervision in doing R&D, and minimize R&D failure rates, because the enterprises' R&D investment has a negative correlation with financial performance, indicating that many technologies or products fail in the development process.

From the characteristics of the samples, enterprises are recommended in the process of collaborative innovation to pay attention to the mutually beneficial relationship with the government, the scientific research institutes, the financial institutions, the educational institutions, and other community development partners. Likewise, there is significant demand for enterprises to establish resource aggregation and utilization mechanisms to manage R&D investment more effectively.

Lastly, this study has some limitations. It explores the influencing channels between different factors and does not consider time as one of the factors; that is, the research applies a cross-sectional design despite the fact that mediation consists of causal processes that unfold over time. The main reason is that the Fourth Industrial Revolution emerged very recently, so the development of the intelligent manufacturing industry started just a few years ago. Currently, there are some difficulties in performing time series analysis because it is too early to get consistent and sufficient data collection.

In addition, this study analyzes the influence on enterprises from the three external factors: economy, policy, and region, whereas there are a few other factors mentioned in this article that will be studied in more detail and depth in the future.

**Author Contributions:** Conceptualization, G.Z.; methodology, G.Z.; software, G.Z.; validation, G.Z.; formal analysis, G.Z.; investigation, G.Z. and Y.L.; resources, G.Z.; data curation, G.Z.; writing—original draft preparation, G.Z.; writing—review and editing, Y.L.; visualization, G.Z.; supervision, Y.L. Both authors have read and agreed to the published version of the manuscript.

**Funding:** This research received no external funding.

**Data Availability Statement:** The datasets generated during and/or analyzed during the current study are not publicly available due to protection of respondent confidentiality but are available from the corresponding author on reasonable request.

**Conflicts of Interest:** The authors declare no conflict of interest.

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
