# Peer review of "Determinants of Financial Performance in China’s Intelligent Manufacturing Industry: Innovation and Liquidity"

_ijfs, doi:10.3390/ijfs9010015_

Round 1

Reviewer 1 Report

See the file attached

Author Response

We appreciate your valuable comments and suggestions greatly. We have made all the revisions in the manuscript which have been highlighted in yellow color.

We attached a file of our concrete responses to your comments and suggestions, please see the attachment.

Thank you! 

Best regards,

Zhang and Lee

Reviewer 2 Report

According to the abstract, this study focuses on '…focuses on the mediation channels through which the financial performance of intelligent manufacturing industries closely related to the Fourth Industrial Revolution has been affected.’ It concluded that ‘…find that innovation efforts mainly led by increasing investment in R&D, along with high liquidity, surely lead to good financial performance, whereas innovation efforts alone do not’

The structure and topic of the study is fit to the requirements and aim of the journal. The paper is characterized by the pursuit of complexity and internal content proportionality. The authors researched with content and methods appropriate to the objectives. The methods used in the empirical analysis and the conclusions drawn from the results are reported accurately.

Overall, in this point of view, the manuscript can be accepted after minor amendments:

Suggestions:

  • Highlight that the selected 317 enterprises which are related to intelligent manufacturing and listed in the China Stock Market is a representative sample or not. What kind of selection method is preferred?
  • The results for the chosen method and variables are limited. The most important is the omitted variable bias, as the variables selected by the authors reflect a few characteristics of the relationship between i.e. financial performance, economic development and innovation. Another issue is that most empirical tests of mediation utilize cross-sectional data despite the fact that mediation consists of causal processes that unfold over time. These bias should be added to the conclusion section.

Author Response

We appreciate your valuable comments and suggestions greatly. We have made all the revisions in the manuscript which have been highlighted in yellow color. 

We attached a file of our concrete responses to your comments and suggestions, please retrieve it.

Thank you!

Best regards,

Zhang and Lee

Round 2

Reviewer 1 Report

The authors have made a great effort to answer the first review and have significantly improved the paper in those parts that were weaker. The article in its current writing meets the standards of a scientific paper in a satisfactory way. Therefore my decision is to accept it in the current form.